# An Online Statistical Framework for Out-of-Distribution Detection

Xinsong Ma[1]   Xin Zou[1]   Weiwei Liu[1]

## Abstract

Out-of-distribution (OOD) detection task is significant in reliable and safety-critical applications. Existing approaches primarily focus on developing the powerful score function, but overlook the design of decision-making rules based on these score function. In contrast to prior studies, we rethink the OOD detection task from an perspective of online multiple hypothesis testing. We then propose a novel generalized LOND (g-LOND) algorithm to solve the above problem. Theoretically, the g-LOND algorithm controls false discovery rate (FDR) at pre-specified level without the consideration for the dependence between the p-values. Furthermore, we prove that the false positive rate (FPR) of the g-LOND algorithm converges to zero in probability based on the generalized Gaussian-like distribution family. Finally, the extensive experimental results verify the effectiveness of g-LOND algorithm for OOD detection.

## 1. Introduction

The ability to detect Out-of-Distribution (OOD) inputs is a fundamental requirement for deploying deep learning models in safety-critical applications (Liang et al., 2018; Sastry & Oore, 2020; Ma et al., 2022). Although DNNs have demonstrated impressive capabilities across a range of complex tasks, such as speech recognition and machine translation (He et al., 2016; Amodei et al., 2016; Dankers et al., 2022), they are inherently vulnerable to OOD examples that deviate from the distribution of training data ( i.e., in-distribution (ID) data ). This vulnerability can lead to overconfident yet incorrect predictions, which is particularly problematic in domains such as computer vision (Sun et al., 2022; Wei et al., 2022).

OOD detection has garnered considerable attention recently and an extensive collection of literature has been developed to tackle this issue (Liu et al., 2020; Yang et al., 2021; Wang et al., 2022; Liu et al., 2023), which mainly devote to obtaining a powerful score function. However, these methods neglect to design a decision rule based on their score function, and directly use traditional threshold-based decision rule, where the threshold is determined such that the true positive rate (TPR) on ID validation set is 95% before testing (Sun et al., 2022; Wei et al., 2022). But the traditional decision rule is empirical and thus its outputs lack rigorous statistical guarantee. Consequently, a natural question arise: *How to design an OOD detection method with rigorous theoretical guarantee and well empirical performance?* This paper aims to systematically study this question.

Different from previous OOD detection literature, we investigate the OOD detection problem from statistical perspective. Note that in some sensitive applications such as financial transactions (Özbayoglu et al., 2020) and self-driving (Huang et al., 2020), the testing examples arrive sequentially in a stream, and the machine learning model must make decisions in time. Hence, we frame the OOD detection task as an online multiple hypothesis testing problem. (Javanmard & Montanari, 2015) proposes the LOND algorithm to address this testing problem, which has been heavily studied in various applications (Javanmard & Montanari, 2015; Robertson & Wason, 2018; Robertson et al., 2022; Liou et al., 2023). Theoretically, to control the false discovery rate (FDR) [2], the LOND algorithm demands that the p-values corresponding to different null hypotheses are mutually independent, which easily leads to a contradiction to real-world applications. We then propose a novel generalized LOND (g-LOND) algorithm to eliminate these restrictions on p-values for FDR control. Moreover, we derive the asymptotic false positive rate (FPR) of the g-LOND algorithm based on the generalized Gaussian-like distribution family. Finally, extensive experiments demonstrate the promising performance of the g-LOND algorithm on OOD detection. We summarize our contributions below:

(1) We frame the OOD detection task as an online multi-

[1]School of Computer Science, National Engineering Research Center for Multimedia Software, Institute of Artificial Intelligence and Hubei Key Laboratory of Multimedia and Network Communication Engineering, Wuhan University, Wuhan, China. Correspondence to: Weiwei Liu <liuweiwei863@gmail.com>.

*Proceedings of the 42nd International Conference on Machine Learning*, Vancouver, Canada. PMLR 267, 2025. Copyright 2025 by the author(s).

---

[2]FDR is related to the concept of *Precision*, and can be considered as the generalization of type-I error for single hypothesis testing. See the motivation of controlling FDR in Section 3.2.

ple hypothesis testing problem and propose a novel g-LOND algorithm to solve it. Our method is distribution-free, and does not rely on the extra information of OOD data.

(2) Theoretically, the g-LOND algorithm enables to control the FDR at a pre-specified level. Besides, we demonstrate that the FPR for the g-LOND algorithm converges to zero in probability based on the generalized Gaussian-like distribution family.

(3) Finally, we conduct extensive experiments to demonstrate the effectiveness of the g-LOND algorithm for OOD detection. The results show that our method enables to achieve a better tradeoff between TPR and FPR compared with previous methods.

## 2. Related Work

**OOD Detection.** Existing efforts on OOD detection mainly focus on classification-based methods, density-based methods and distance-based methods. The classification-based methods directly obtain confidence from the classifier which integrates some specific modifications for loss functions or classifier architectures (Lee et al., 2018; Liu et al., 2020; Huang et al., 2021; Liu et al., 2023; Djurisic et al., 2023) with some beneficial design to OOD detection. These designs mainly focus on the loss function (Liu et al., 2020; Huang et al., 2021; Liu et al., 2023), classifier architecture (Lee et al., 2018; Djurisic et al., 2023), and some post-hoc processing techniques (Hendrycks & Gimpel, 2017; Liang et al., 2018). The density-based methods characterize the in-distribution distribution using probabilistic models (Abati et al., 2019; Ren et al., 2019). The distance-based methods usually compute distance in the high-dimension space such as feature space and gradient space to distinguish ID and OOD example (Hendrycks et al., 2022; Sun et al., 2022). Additionally, minor methods involve introducing some OOD samples to train the new networks (Yang et al., 2021; Ming et al., 2022). The above methods focus on designing or training a powerful score function and then apply empirical decision rule to identify the OOD examples directly. However, this decision rule is empirical, thus the outputs of it lack rigorous theoretical guarantee. This paper aims to solve this problem.

**FDR Control** Similar to control type-I error in single hypothesis testing, how to control FDR is the core problem for multiple hypothesis testing algorithms (Benjamini & Hochberg, 1995; Blanchard & Roquain, 2008; Yu et al., 2023; Ma et al., 2024; 2025). Online multiple hypothesis testing is first introduced by (Foster & Stine, 2008), who propose a alpha investing method to control mFDR for the streamed variable selection problem in high dimensional space. Many following researches (Aharoni & Rosset, 2014;

Javanmard & Montanari, 2018) develop this work and make this type of methods more powerful. (Javanmard & Montanari, 2015) propose the LOND and LORD algorithms to control FDR and mFDR. The LOND algorithm has been heavily studied in various applications (Javanmard & Montanari, 2015; Robertson & Wason, 2018; Robertson et al., 2022; Liou et al., 2023). Note that these works mainly focus on the situation where p-values are mutually independent or satisfy the special dependence such as conditional superuniformity. Unlike these literature, this paper aims to propose a novel algorithm to control FDR without the consideration of dependence between p-values.

## 3. Preliminaries

### 3.1. Out-of-Distribution Detection

We donote by $\mathcal{X} \subseteq \mathbb{R}^d$ the feature space and $\mathcal{Y} = \{1, 2, 3, \ldots, K\}$ the label space with the joint distribution $\mathcal{D}$ on $\mathcal{Z} = \mathcal{X} \times \mathcal{Y}$. Besides, $\mathcal{X}$ has marginal distribution $\mathcal{D}_{in}$. Let $(x, y)$ be the feature-label pair, where instance $x \in \mathcal{X}$ and label $y \in \mathcal{Y}$. We denote the transpose of vector/matrix by the superscript $'$. Denote by $f_\theta : \mathcal{X} \to \mathcal{Y}$ the classifier (or hypothesis) which is parameterized by $\theta \in \Theta$, where $\Theta$ is the parameter space and maps $x$ to a certain label $y$. In addition, we denote by $\mathcal{L}(f_\theta(x), y)$ the certain loss function.

In the canonical classification setting, the primary goal is to maximize the standard accuracy of models on the unseen examples from the underlying distribution $\mathcal{D}_x$. Concretely, we hope to find the classifier $\hat{f}(\cdot)$ with the smallest risk:

$$R(\hat{f}) = \min_{\theta \in \Theta} \mathbb{E}_{(x,y) \sim \mathcal{D}} \mathcal{L}(f_\theta(x), y).$$

During the prediction phase, the test examples are usually assumed to come from the same distribution $\mathcal{D}_{in}$ as the training set. Nevertheless, practical situations may introduce some inputs from unfamiliar distributions, with label space potentially lacking any intersection with $\mathcal{Y}$. These inputs are referred to as the OOD data and should not be predicted.

The goal of OOD detection is to identify the OOD examples in testing set. In the previous literature, the OOD detection task is formulated as the following binary decision problem:

$$\phi(x) = \begin{cases} ID, & \text{if } s(x) \geq s^* \\ OOD, & \text{if } s(x) < s^* \end{cases} \tag{1}$$

where $s(\cdot)$ denotes the score function and the threshold $s^*$ is empirically selected so that the ture positive rate (TPR) on ID validation set is $95\%$ before testing (Sun et al., 2022; Wei et al., 2022).

The previous studies about OOD detection mainly devote to obtaining a powerful score function which captures the

discriminative information in ID data (Hendrycks & Gimpel, 2017; Liu et al., 2020; Djurisic et al., 2023; Liu et al., 2023). However, the question of how to make decisions based the score functions is under-explored for online OOD detection. In contrast to the previous literature, we investigate the OOD detection problem starting from the decision rule.

### 3.2. A Statistical Framework for OOD detection

Unlike the existing literature, we provide a new insight for the OOD detection problem from an online hypothesis testing perspective . Specifically, for a testing set $\mathcal{T}^{test} = \{X_1^{test}, X_2^{test}, \cdots\}$, OOD detection task is formulated as the following hypothesis testing problem:

$$
\begin{aligned}
H_{1;0} &: X_1^{test} \sim \mathcal{D}_{in}, \quad H_{1;1} : X_1^{test} \not\sim \mathcal{D}_{in} \\
H_{2;0} &: X_2^{test} \sim \mathcal{D}_{in}, \quad H_{2;1} : X_2^{test} \not\sim \mathcal{D}_{in} \\
&\qquad\qquad \vdots \\
H_{n;0} &: X_n^{test} \sim \mathcal{D}_{in} \quad H_{n;1} : X_n^{test} \not\sim \mathcal{D}_{in} \\
&\qquad\qquad \vdots
\end{aligned} \tag{2}
$$

where $H_{i;0}$ and $H_{i;1}$ are called null hypothesis and alternative/non-null hypothesis, respectively. If $H_{i,0}$ is rejected, we declare that $X_i^{test}$ is OOD.

To support or reject the claim in null hypothesis, commonly we use the concept of *p-value* as the measure of statistical significance. Its definition is as follows.

**Definition 3.1.** [p-value (Casella & Berger, 2001)] Given a sample $\widetilde{X}$ [3]. A statistic $p(\widetilde{X})$ is called p-value corresponding to the null hypothesis $H_0$, if $p(\widetilde{X})$ satisfies

$$
\mathbb{P}[p(\widetilde{X}) \leq t | H_0] \leq t \tag{3}
$$

for every $0 \leq t \leq 1$.

Obviously, if $p(\widetilde{X})$ follows the uniform distribution on $(0, 1)$ under the null hypothesis, then $p(\widetilde{X})$ is a valid p-value. A small p-value usually provides strong evidence against the null hypothesis. It is noteworthy that the p-value has clear statistical interpretation. For example, suppose the p-value of a OOD testing example $X_i^{test}$ is 0.01. This means that for any coming testing example $X_j^{test}$, the probability that $X_j^{test}$ is more similar to OOD data than $X_i^{test}$ is 0.01. In other words, it is extremely difficult to find a example that is less like OOD data than $X_i^{test}$. Hence, we are highly confident that $X_i^{test}$ is OOD.

*Remark* 3.2. In statistics, the following terminology characterizes the distribution of null p-values: if $\mathcal{P}[p(\widetilde{X}) \leq t | H_0] = t$, the p-value $p(\widetilde{X})$ is called exact or uniform; if $\mathcal{P}[p(\widetilde{X}) \leq t | H_0] < t$, $p(\widetilde{X})$ is called conservative. Compared to an exact p-value, a conservative one tends to understate the evidence against the null Hypothesis.

---

[3]A sample means a sequence of examples.

In single hypothesis testing, if the p-value is below a prescribed significance level $\alpha$, then the null hypothesis is rejected. While this ritual enables to control the probability of type-I error, in the case of multiple hypotheses testing we need to adjust the significance levels to control other metrics, in which false discovery rate (FDR) proposed by (Benjamini & Hochberg, 1995) is widely used. The statistical advantages of FDR have been detailedly discussed in (Benjamini & Hochberg, 1995; Benjamini & Yekutieli, 2001).

Given a sequence of the null hypotheses $\{H_{i;0}\}_{i \geq 1}$, we let $\mathcal{R}(n)$ be the set of indices of the rejected hypotheses for the first $n$ null hypotheses $\{H_{1;0}, H_{2;0}, \cdots, H_{n;0}\}$. Similarly, denote by $\mathcal{H}_0(n)$ and $\mathcal{H}_1(n)$ the set of indices for the true null hypotheses and false null hypothesis for $\{H_{1;0}, H_{2;0}, \cdots, H_{n;0}\}$, respectively. Let $n_0 = |\mathcal{H}_0(n)|$ be the number of true null hypotheses. In statistics, if one null hypothesis is rejected, it is said to make a discovery. FDR is the expected proportion of false discoveries among the rejected hypotheses.

**Definition 3.3** (FDR(Benjamini & Hochberg, 1995)). Let $V$ denote the number of true null hypotheses rejected; moreover, let $R$ be the number of rejected hypotheses, the false discovery proportion (FDP) for the first $n$ null hypotheses is defined as:

$$
\mathrm{FDP}(n) = \begin{cases} V/R, & \text{if } R > 0, \\ 0, & \text{otherwise.} \end{cases}
$$

The expectation of $\mathrm{FDP}(n)$ is called the FDR, namely

$$
\mathrm{FDR}(n) = \mathbb{E}(\mathrm{FDP}(n)) = \mathbb{E}\left[\frac{|\mathcal{R}(n) \cap \mathcal{H}_0(n)|}{\max\{1, |\mathcal{R}(n)|\}}\right].
$$

The FDR can be considered as the generalization of the probability of type-I error for multiple hypothesis testing. Therefore, multiple testing algorithms should control its FDR first. According to Ma et al. (2024), the FDP is closely related with the concept of *Precision*. Denote by $\mathcal{R}^c(n)$ the complement of set $\mathcal{R}(n)$. Using the confusion matrix notations [4], the FDP can be also expressed as

$$
\mathrm{FDP}(n) = \frac{FN}{FN + TN}.
$$

The *Precision* is defined as

$$
\text{Precision} = \frac{TP}{TP + FP} = \frac{|\mathcal{R}^c(n) \cap \mathcal{H}_0(n)|}{\max\{|\mathcal{R}^c(n)|, 1\}}.
$$

Thus, the FDP is the "dual" concept of the *Precision*.

---

[4]Based on the notations of $\mathcal{R}(n)$, $\mathcal{H}_0(n)$ and $\mathcal{H}_1(n)$, we have the following relations: $TP = |\mathcal{R}^c(n) \cap \mathcal{H}_0(n)|$, $FN = |\mathcal{R}(n) \cap \mathcal{H}_0(n)|$, $FP = |\mathcal{R}^c(n) \cap \mathcal{H}_1(n)|$ and $TN = |\mathcal{R}(n) \cap \mathcal{H}_1(n)|$.

### 3.3. LOND Algorithm

To control FDR for the testing problem in (2), a direct idea is to apply the LOND algorithm, which has been heavily studied in various applications (Javanmard & Montanari, 2015; Robertson & Wason, 2018; Robertson et al., 2022; Liou et al., 2023).

**Definition 3.4** (LOND (Javanmard & Montanari, 2015))**.** Given a sequence of null hypotheses $H_{1,0}, H_{2,0}, \cdots$ which arrive sequentially in a stream, with corresponding p-values $p_1, p_2, \cdots$. For a prescribed level $\alpha$, choose any sequence of positive numbers $\boldsymbol{\gamma} = \{\gamma_i\}_{i=1}^{\infty}$ such that $\sum_{i=1}^{\infty} \gamma_i = 1$ and define

$$\alpha_i = \alpha \gamma_i (D(i-1) + 1),$$

where $D(0) = 0$ and $D(i) = \sum_{k=1}^{i} \mathbf{1}(p_k \leq \alpha_k)$. Then, the null hypothesis $H_{i,0}$ is rejected if its corresponding p-value $p_i$ satisfies $p_i \leq \alpha_i$.

In statistics, $\alpha$ is commonly specified as 0.05. Similar to the type-I error, in multiple hypothesis testing, a testing algorithm for problem 2 should make as many discoveries as possible (low FPR) while maintaining the FDR at a prescribed level. Javanmard & Montanari (2015) prove that if p-values corresponding to all null hypotheses are available and mutually independent, then LOND algorithm controls FDR at pre-specified level.

In most multiple hypothesis testing literature (Benjamini & Hochberg, 1995; Benjamini & Yekutieli, 2001; Blanchard & Roquain, 2008; Delattre & Roquain, 2015; Cao et al., 2022), the p-values or the distribution of the testing statistic are assumed to be known. Denote by $F(\cdot)$ the cumulative distribution function of $s(X)$ where $s(\cdot)$ is the score function and $X \sim \mathcal{D}_x$. Then, for a given example $X^{test}$, its p-value can be expressed as

$$
\begin{aligned}
p(X^{test}) &= \mathbb{P}_{X \sim \mathcal{D}_x}(s(X) \leq s(X^{test})) \\
&= F(s(X^{test})).
\end{aligned}
\tag{4}
$$

According to the Definition 2, under the $H_0$ ($X^{test}$ is the ID data), we have

$$
\begin{aligned}
\mathbb{P}\left(F(s(X^{test})) \leq x\right) &= \mathbb{P}\left(s(X^{test}) \leq F^{-1}(x)\right) \\
&= F(F^{-1}(x)) = x,
\end{aligned}
$$

where $F^{-1}(\cdot)$ is the inverse function of $F(\cdot)$. Therefore, the random variable $F(s(X^{test}))$ follows the uniform distribution on (0, 1), namely, $p(X^{test})$ is a valid p-value and is exact. Obviously, small score $s(X^{test})$ leads to a small p-value, consistent with the classical setting of OOD detection in Eq.(1) and the meaning of the p-value.

However, in the context of the OOD detection, often we have litle prior information about underlying distribution $F(\cdot)$. Hence, the empirical p-value is often used in real-world

applications, which is a nonparametric estimation method for the p-value $p(X^{test})$. In statistics, let $X_1, X_2, \cdots, X_m$ be a sample of observations (i.e., random variable). For testing example $X$, the empirical p-value of $X$ is defined as

$$\hat{p}(X) = \frac{\sum_{j=1}^{m} \mathbb{1}(X_j \leq X) + 1}{m + 1}.$$

Note that $X$ is one-dimensional random variable. Nevertheless, in OOD detection, the inputs are high-dimensional images. Therefore, utilizing the score function designed for OOD detection, we reduce each high-dimensional images to the univariate score, and then compute the empirical p-values. Specifically, given a calibrated set $\mathcal{T}^{cal} = \{X_1^{cal}, X_2^{cal}, \ldots, X_m^{cal}\}$ consisting of ID data, for a testing example $X_i^{test}$, the empirical p-value $p_i$ corresponding to null hypothesis $H_{i;0}$ is expressed as

$$p_i = p(X_i^{test}) = \frac{|\{j \in [m] : \hat{s}(X_j^{cal}) \leq \hat{s}(X_i^{test})\}| + 1}{m + 1},$$
$$\tag{5}$$

where $\hat{s}(\cdot)$ is a certain score function. According to (Arlot et al., 2010), we can easily verify that empirical p-value in Eq. (5) satisfies Definition 3.1. Note that the empirical p-values for different testing examples are not independent since they rely on a common calibrated set $\mathcal{T}^{cal}$. Hence, we cannot directly apply LOND algorithm with empirical p-values to control FDR.

### 3.4. Generalized LOND Algorithm

To control FDR, we modify the LOND algorithm and then propose a novel generalized LOND (g-LOND) algorithm to remove the constraints on the dependence between p-values. We denote $f_+(0) = \lim_{x \to 0+} f(x)$, and define two function classes:

$$\mathcal{F}_1 = \{f(x) : f_+(0) = 0, f'(x) \geq 1\},$$
$$\mathcal{F}_2 = \{f(x) : f_+(0) = 0, f'(x) > 0, \int_0^1 \frac{1}{f(x)} \, dx \leq 1\}$$

for $x \in (0, 1)$. It is easy to verify that $\frac{-1}{\log x} \in \mathcal{F}_2$. The g-LOND algorithm is defined as follows.

**Definition 3.5** (g-LOND)**.** Given a sequence of null hypotheses $H_{1,0}, H_{2,0}, \cdots$ which arrive sequentially in a stream, with corresponding p-values $p_1, p_2, \cdots$. For a prescribed level $\alpha$, choose a function $f(\cdot) \in \mathcal{F}_1 \cup \mathcal{F}_2$ and a sequence of positive numbers $\boldsymbol{\gamma} = \{\gamma_i\}_{i=1}^{\infty}$ such that $\sum_{i=1}^{\infty} \gamma_i = 1$, define

$$\alpha_i = \alpha \gamma_i (D(i-1) + 1),$$

where $D(0) = 0$ and $D(i) = \sum_{k=1}^{i} \mathbf{1}(f(p_k) \leq \alpha_k)$. Then, the null hypothesis $H_{i,0}$ is rejected if its corresponding p-value $p_i$ satisfies $f(p_i) \leq \alpha_i$.

**Algorithm 1** Practical g-LOND Algorithm

1: **Input:** Training set $\mathcal{T}$, calibrated set $\mathcal{T}^{cal} = \{X_1^{cal}, X_2^{cal}, \ldots, X_m^{cal}\}$ testing set $\mathcal{T}^{test} = \{X_1^{test}, X_2^{test}, \ldots, X_n^{test}, \cdots\}$, prescribed level $\alpha \in (0, 1)$ and sequence $\{\gamma_i\}_{i=1}^{\infty}$.

2: Train the score function on $\mathcal{T}$:

$$\hat{s}(x) = \max_{i \in [K]} \frac{f_i}{\|\boldsymbol{f}\|},$$

where $\boldsymbol{f}$ is the logit Output.

3: Calculate the p-value corresponding to $X_i^{test}$:

$$p_i = \frac{|\{j \in [m] : \hat{s}(X_j^{cal}) \le \hat{s}(X_i^{test})\}| + 1}{m + 1}.$$

4: Compute the significance level corresponding to $X_i^{test}$:

$$\alpha_i = \alpha\gamma_i(D(i-1)+1),$$

5: **Output:** Declare that $X_i^{test}$ is OOD if $f(p_i) \le \alpha_i$.

The practical g-LOND algorithm with empirical p-values is presented in Algorithm 3.4. Inspired by (Hendrycks et al., 2022), we use maximum normalied logit $\hat{s}(x) = \max_{i \in [K]} \frac{g_i}{\|\boldsymbol{g}\|}$ as our score function.

## 4. FDR Control of g-LOND Algorithm

We first investigate the theoretical properties of the g-LOND algorithm about FDR. It is well known that if p-values $p_1, p_2, \ldots, p_n$ are mutually independent or PRDS, the BH procedure can control FDR at level $\alpha$ (Benjamini & Hochberg, 1995; Benjamini & Yekutieli, 2001). Factually, the g-LOND algorithm also enjoys this theoretical result. We begin with the following definitions.

**Definition 4.1 (Increasing Set).** A subset $\mathcal{A} \subset \mathbb{R}^n$ is said to be increasing if for all $x \in \mathcal{A}$, $x \le y$ implies $y \in \mathcal{A}$, where the comparison of $x$ and $y$ is component-wise.

Then, PRDS property is defined as follows.

**Definition 4.2 (PRDS**(Benjamini & Yekutieli, 2001)**).** A family of random variables $\{X_1, X_2, \ldots, X_n\}$ is said to be PRDS on a subset $I_0 \subset \{1, 2, \ldots, n\}$ if for all $i \in I_0$, the function $\mathbb{P}((X_1, X_2, \ldots, X_n) \in \mathcal{D}|X_i = x)$ is an increasing function in $x$ for any increasing subset $\mathcal{A}$.

To derive our theoretical results, we first introduce the following lemma.

**Lemma 4.3.** *((Ma et al., 2024)) Suppose that the p-values $p_1, p_2, \ldots, p_n$ are PRDS on $\mathcal{H}_0(n)$.*

1. *Denote $p_i^* := f(p_i)$ for all $i \in \{1, 2, \ldots, n\}$ where $f(\cdot)$ is strictly increasing or decreasing. Then*

$\{p_1^*, p_2^*, \ldots, p_n^*\}$ *is PRDS on $\mathcal{H}_0(n)$ as well.*

2. *For any $i \in \mathcal{H}_0(n)$, the function*

$$\mathbb{P}((p_1, p_2, \ldots, p_n) \in \mathcal{A} \mid p_i \le x)$$

*is increasing in $x$ for any increasing set $\mathcal{A}$.*

Lemme 4.3 indicates that the PRDS is invariant for monotonic transformation. Then, our core theorem is presented as follows.

**Theorem 4.4.** *Given a prescribed level $\alpha \in (0, 1)$, a sequence of p-values $p_1, p_2, \cdots$ corresponding to $H_{1,0}, H_{2,0}, \cdots$ and $f(\cdot) \in \mathcal{F}_1$. If $p_1, p_2, \ldots, p_n$ are mutually independent or PRDS, then the FDR of g-LOND algorithm satisfies*

$$\text{FDR}_{g-LOND} \le \alpha.$$

The proof of Theorem 4.4 is presented in Appendix A.1. Obviously, the empirical p-values are not independent, since they depend on the same trained socre function and calibrated set. (Yu et al., 2023) demonstrates that the empirical p-values are conditionally PRDS under some assumptions. This paper aims to conduct an in-depth study of the g-LOND algorithm under weaker conditions for FDR control. Then, we have following theoretical result.

**Theorem 4.5.** *Given a prescribed level $\alpha \in (0, 1)$, a sequence of empirical p-values $p_1, p_2, \cdots$ corresponding to $H_{1,0}, H_{2,0}, \cdots$ and $f(\cdot) \in \mathcal{F}_2$. Then the FDR of g-LOND algorithm satisfies*

$$\text{FDR}_{g-LOND} \le \alpha.$$

The proof of Theorem 4.5 is presented at Appendix A.2. Theorem 4.5 indicates that the g-LOND algorithm controls FDR at a prescribed level regardless of the dependence between the empirical p-values. Obviously, such a argument is invalid for many popular methods such as LOND algorithm and LORD algorithm. which broadens the applicability scope of the g-LOND algorithm.

*Remark* 4.6. Motivated by (Benjamini & Yekutieli, 2001), (Javanmard & Montanari, 2015) show that the LOND algorithm can also control FDR for dependent p-values by choosing special sequence $\hat{\gamma}$ where

$$\hat{\gamma}_i = \frac{\gamma_i}{\sum_{k=1}^{i} \frac{1}{k}}.$$

Obviously, such a algorithm is too conservative and almost accepts all null hypotheses for large $i$ [5], resulting in a high FPR.

---

[5]Note that $\sum_{k=1}^{i} \frac{1}{k}$ is monotonically increasing w.r.t. $i$ and tends to $\infty$ as $i \to \infty$. Therefore, for large $i$, we have $\hat{\alpha}_i = \alpha\hat{\gamma}_i(D(i-1)+1) \to 0$.

# 5. Asymptotic FPR of g-LOND Algorithm

False positive rate (FPR) is a significant evaluation criterion for OOD detection. Using the notations $\mathcal{R}$ and $\mathcal{H}_1$, the FPR can be expressed as

$$\text{FPR} = \frac{FP}{FP + TN} = \frac{|\mathcal{R}^c(n) \cap \mathcal{H}_1(n)|}{|\mathcal{H}_1(n)|}.$$

Although FDR control has been widely studied, relatively little is known about the theoretical properties of FPR. In this section, we investigate the asymptotic behavior of FPR for the g-LOND algorithm. Our analytical framework is similar to that of (Donoho & Jin, 2004; Neuvial & Roquain, 2012).

## 5.1. Analytical Framework

In the field of multiple hypothesis testing, many theoretical studies (Javanmard & Montanari, 2015; 2018) assume that p-values are available, equivalently, the distribution of test statistic for each hypothesis is known. In this case, p-values can be expressed as $p_i = \Psi(T_i)$, where $\Psi(\cdot)$ is the survival function of test statistic and $T_i$ is the observation of test statistic corresponding to $H_i$. For mathematical convenience, we impose that $T_1, T_2, \ldots, T_n$ are independent continuous random variables. Reasonably, working with p-values $p_1, p_2, \ldots, p_n$ is equivalent to working with observations $T_1, T_2, \ldots, T_n$. We first define a function class as follows.

$$\mathcal{F} = \left\{ \Psi(t) : \lim_{t \to \infty} \frac{-\log \Psi(t)}{t^\lambda} = \frac{1}{\lambda}, \ \lambda > 1 \right\}.$$

In this section, we describe the distribution of the observation $T_i$ in terms of the generalized Gaussian-like family, which is a variant of the generalized Gaussian distribution.

**Definition 5.1** (Generalized Gaussian-like Distribution Family). A random variable $X$ is said to follow the generalized Gaussian-like distribution family with the location $\mu$ and the degree $\lambda$, denote $X \sim G(\mu, \lambda)$, if the survival function $\Psi(\cdot)$ of $X$ satisfies $\Psi(t - \mu) \in \mathcal{F}$.

It is easy to verify that the popular Gaussian distribution and Laplace distribution belong to the generalized Gaussian-like distribution family, which are often used in the theoretical analysis.

Our inspiration for considering the tail generalized Gaussian distribution comes from the previous works (Donoho & Jin, 2004; Ingster & Suslina, 2012) on global testing. In terms of the notation $G(\mu, \lambda)$, we assume that the observation $T_i$ is distributed as

$$T_i \sim \begin{cases} G(0, \lambda) & \text{if } i \in \mathcal{H}_0(n) \\ G(\mu, \lambda) & \text{if } i \in \mathcal{H}_1(n), \end{cases} \quad (6)$$

where $\mu > 0$ is allowed to vary with the number of hypotheses $n$. Eq. (6) shows that the non-null hypothesis is distinguished frome null hypothesis by a positive mean shift $\mu$. In (Donoho & Jin, 2004), $\mu$ is set to $\sqrt{2r \log(n)}$. In this section, $\mu$ is parameterized as

$$\mu = \left( \lambda r \log n \right)^{1/\lambda} \quad (7)$$

where $r > 0$. Following (Donoho & Jin, 2004; 2006; Jin & Ke, 2016), we focus on the sparse region in which the number of true null hypothesis is larger than that of the true alternative hypothesis. In this case, we have

$$n_1 = |\mathcal{H}_1(n)| = n^{1-\beta},$$

where $\beta < r < 1$. Besides, we denote

$$\epsilon = \frac{n_1}{n} = n^{-\beta}.$$

## 5.2. Asymptotic Property of FPR for g-LOND Algorithm

For simplicity, let $\gamma_i = \frac{C}{i^v}$ for $v > 1$, where

$$\frac{1}{C} = \sum_{i=1}^{\infty} \frac{1}{i^v}.$$

Denote $\tau_k$ as the time of $k$-th rejection (with $\tau_0 = 0$). Based on the analytical framework in Section 5.1, we first establish the upper bound of $\tau_k$ expectation [6].

**Theorem 5.2.** *Given the first $n$ hypotheses $H_1, H_2, \cdots, H_n$, and suppose that the corresponding observations $T_1, T_2, \ldots, T_n$ satisfy the condition (6). Then, the expectation of $\tau_k \wedge n_1$ satisfies*

$$\mathbb{E}(\tau_k \wedge n_1) \leq 3k \cdot n_1^\beta \quad \text{for all } 0 < k < n,$$

*where $n_1 = |\mathcal{H}_1|$.*

The proof of Theorem 5.2 is presented at Appendix A.3. Theorem 5.2 shows that the number of rejections is impacted by the sequence $\{\gamma_i\}_{i=1}^{\infty}$ and the parameter $\beta$. For example, small $\beta$ leads to small $\mathbb{E}(\tau_k \wedge n_1)$, meaning that the g-LOND algorithm tends to make more rejections. Based on Theorem 5.2, we obtain the following core result.

**Theorem 5.3.** *Given the first $n$ hypotheses $H_1, H_2, \cdots, H_n$, and suppose that the corresponding observations $T_1, T_2, \ldots, T_n$ satisfy the condition (6) with $\Psi(0) = \frac{1}{2}$. If $\beta < \frac{1}{2}$ and $r > \beta + +v - 1$, then the FPR of g-LOND algorithm satisfies*

$$\frac{FP}{FP + TN} \to 0 \quad \text{in probability}.$$

The proof of Theorem 5.3 is presented at Appendix A.4. Theorems 4.5 and 5.3 show that the FPR of g-LOND algorithm tends to zero in probability while maintaining its FDR at the prescribed level.

---

[6] $a \wedge b$ means $\min\{a, b\}$.

*Table 1.* The experimental results on **CIFAR-100** as ID data. We use ResNet18 as the backbone for each method. The best results are highlighted in **bold**.

| Data | CIFAR-10 | | | TinyImageNet | | | SVHN | | | Texture | | | Place365 | | |
|---|---|---|---|---|---|---|---|---|---|---|---|---|---|---|---|
| Practical | TPR | FPR | F1 | TPR | FPR | F1 | TPR | FPR | F1 | TPR | FPR | F1 | TPR | FPR | F1 |
| ASH | 94.46 | 78.38 | 67.10 | 94.46 | 73.92 | 76.16 | 94.46 | 63.47 | 50.07 | 94.46 | 72.40 | 78.78 | 94.46 | 77.58 | 38.91 |
| Cider | 93.27 | 86.80 | 64.39 | 93.27 | 74.16 | 75.51 | 93.27 | 42.80 | 61.00 | 93.27 | 70.30 | 78.60 | 93.27 | 76.63 | 38.79 |
| Energy | 94.52 | 78.83 | 67.01 | 94.52 | 72.15 | 76.58 | 94.52 | 74.79 | 46.06 | 94.52 | 78.85 | 77.50 | 94.52 | 78.83 | 38.56 |
| KLM | 80.56 | 56.06 | 65.53 | 80.56 | 57.54 | 71.55 | 80.56 | 43.77 | 52.53 | 80.56 | 53.57 | 77.51 | 80.56 | 45.63 | 42.85 |
| MSP | 94.68 | 79.28 | 66.96 | 94.68 | 73.58 | 76.34 | 94.68 | 78.69 | 44.84 | 94.68 | 80.24 | 77.30 | 94.68 | 78.69 | 38.65 |
| RankFeat | **95.15** | 95.26 | 63.23 | **95.15** | 91.77 | 72.72 | **95.15** | 92.94 | 41.02 | **95.15** | 89.87 | 75.68 | **95.15** | 94.00 | 34.73 |
| SHE | 94.38 | 78.23 | 67.10 | 94.38 | 76.37 | 75.58 | 94.38 | 69.00 | 48.07 | 94.38 | 80.04 | 77.19 | 94.38 | 80.40 | 38.05 |
| VIM | 94.15 | 85.82 | 65.04 | 94.15 | 81.77 | 74.30 | 94.15 | 71.29 | 47.26 | 94.15 | 57.50 | 81.80 | 94.15 | 81.84 | 37.57 |
| DICE | 93.95 | 77.78 | 67.02 | 93.95 | 75.55 | 75.54 | 93.95 | 64.88 | 49.33 | 93.95 | 76.43 | 77.70 | 93.95 | 78.92 | 38.35 |
| PALM | 94.55 | 78.29 | 67.17 | 94.55 | 72.23 | 76.58 | 94.55 | 75.10 | 45.94 | 94.55 | 79.20 | 77.44 | 94.55 | 77.52 | 38.96 |
| Gram | 94.61 | 96.67 | 62.65 | 94.61 | 93.23 | 72.16 | 94.61 | 35.54 | 63.71 | 94.61 | 70.47 | 79.25 | 94.61 | 96.32 | 34.03 |
| KNN | 94.61 | 79.32 | 66.92 | 94.61 | 71.67 | 76.74 | 94.61 | 63.47 | 50.05 | 94.61 | 64.64 | 77.48 | 94.61 | 76.02 | 39.43 |
| ODIN | 94.65 | 80.20 | 66.71 | 94.65 | 74.17 | 76.20 | 94.65 | 63.47 | 42.65 | 94.65 | 74.37 | 78.46 | 94.65 | 77.79 | 38.91 |
| g-LOND | 92.4 | **44.19** | **68.08** | 91.95 | **53.77** | **78.36** | 92.08 | **29.76** | **69.94** | 91.69 | **44.82** | **81.8** | 92.83 | **30.62** | **52.87** |
| Classical | FPR95 | AUC | AUPR | FPR95 | AUC | AUPR | FPR95 | AUC | AUPR | FPR95 | AUC | AUPR | FPR95 | AUC | AUPR |
| ASH | 80.06 | 76.48 | 75.66 | 75.80 | 79.92 | 84.69 | 65.89 | 85.60 | 75.89 | 74.36 | 80.72 | 87.10 | 79.42 | 78.76 | 56.57 |
| Cider | 90.49 | 65.68 | 63.26 | 80.32 | 77.26 | 83.43 | 65.52 | 72.91 | 64.34 | 74.70 | 78.23 | 86.45 | 81.84 | 73.37 | 49.24 |
| Energy | 80.40 | 79.05 | 79.75 | 74.09 | 82.76 | 87.96 | 76.66 | 82.03 | 69.84 | 80.14 | 78.35 | 85.79 | 80.46 | 79.52 | 60.62 |
| KLM | 84.26 | 73.91 | 68.97 | 77.80 | 79.22 | 83.02 | 78.43 | 79.34 | 66.69 | 76.28 | 75.77 | 81.21 | 77.35 | 75.70 | 42.46 |
| MSP | 80.36 | 78.47 | 79.60 | 74.63 | 82.07 | 87.81 | 79.66 | 78.42 | 64.92 | 81.26 | 77.32 | 85.42 | 79.64 | 79.22 | 61.13 |
| RankFeat | 95.01 | 58.04 | 60.17 | 91.37 | 65.72 | 74.57 | 92.50 | 72.14 | 62.89 | 89.64 | 69.40 | 81.18 | 93.76 | 63.82 | 40.54 |
| SHE | 80.60 | 78.15 | 78.98 | 78.97 | 79.74 | 85.71 | 71.54 | 80.97 | 65.99 | 82.00 | 73.64 | 81.42 | 82.63 | 76.30 | 54.20 |
| VIM | 87.70 | 72.21 | 72.88 | 83.85 | 77.76 | 85.21 | 74.28 | 83.14 | 74.22 | 60.00 | **85.91** | **91.22** | 83.89 | 75.85 | 56.35 |
| DICE | 81.36 | 78.04 | 78.74 | 79.69 | 80.72 | 86.65 | 69.35 | 84.22 | 72.94 | 79.44 | 77.63 | 85.07 | 82.52 | 78.33 | 59.08 |
| PALM | 79.97 | **79.38** | 80.09 | 74.18 | 83.25 | 88.47 | 76.98 | 81.41 | 76.80 | 80.74 | 78.74 | 86.25 | 79.15 | 80.28 | 62.10 |
| Gram | 96.99 | 49.41 | 49.19 | 93.71 | 53.91 | 62.23 | 22.92 | 95.55 | 90.06 | 71.16 | 70.79 | 74.62 | 96.60 | 46.38 | 20.21 |
| KNN | 80.68 | 77.02 | 74.63 | 73.23 | 82.54 | 88.39 | 64.91 | 84.15 | 72.22 | 66.02 | 83.66 | 89.46 | 77.27 | 79.43 | 58.57 |
| ODIN | 81.31 | 78.18 | 78.88 | 75.61 | 81.63 | 86.89 | 87.16 | 74.54 | 58.41 | 75.59 | 79.33 | 86.23 | 79.05 | 79.45 | 59.24 |
| g-LOND | **72.1** | 78.92 | **80.18** | **66.61** | 83.29 | **88.86** | **54.74** | 84.12 | **76.77** | **56.29** | 84.29 | **89.7** | **68.21** | 81.59 | **65.67** |

## 6. Experiment

In this section, we aim to validate the effectiveness of our proposed g-LOND algorithm. The evaluation criteria include the practical metrics TPR, FPR and F1-score, and the classical metrics FPR95, AUROC and AUPR. Extensive experimental results demonstrate the superiority of our method.

### 6.1. Experimental Settings

We mainly follow the experimental settings in (Yang et al., 2022; Zhang et al., 2023b), and our codes are based on (Zhang et al., 2023b). We next introduce some necessary settings in our experiments.

**Baselines.** We choose some popular OOD detection methods as our baselines, including MSP (Hendrycks & Gimpel, 2017), ODIN (Liang et al., 2018), Gram (Sastry & Oore, 2020), Energy (Liu et al., 2020), VIM (Wang et al., 2022), KNN (Sun et al., 2022), KLM (Hendrycks et al., 2022), RankFeat (Song et al., 2022), DICE (Sun & Li, 2022), ASH (Djurisic et al., 2023), Cider (Ming et al., 2023), SHE (Zhang et al., 2023a) and PALM (Lu et al., 2024).

**Benchmarks.** We use CIFAR-100 (Krizhevsky, 2009) and ImageNet-200 (Deng et al., 2009) as ID data. For CIFAR-100, we use CIFAR-10, TinyImageNet (Krizhevsky et al., 2017), SVHN (Netzer et al., 2011), Texture (Kylberg,

2011), and Places365 (Zhou et al., 2018) as OOD data. For ImageNet-200, we use SSB-hard (Vaze et al., 2022; Zhang et al., 2023b), NINCO (Bitterwolf et al., 2023), iNaturalist (Horn et al., 2018), Textures (Cimpoi et al., 2014), and OpenImage-O (Wang et al., 2022) as OOD data.

**Metrics.** In this paper, we report the practical and classical metrics. The practical metrics include TPR, FPR and F1-score. The classical metrics include FPR95, AUROC and AUPR. In this paper, we regard ID as positive [7].

**Model.** When the CIFAR-100 is used as the ID data, we use the ResNet10 as the backbone for each method. When ImageNet200 is used as ID data, we use the ResNet50 as the backbone for each method. We follow the experimental implementation in Yang et al. (2022); Zhang et al. (2023b). More details can be found in Yang et al. (2022); Zhang et al. (2023b).

### 6.2. Experimental Results

The experimental results on CIFAR-100 as ID data are presented in Tables 1 and the Results on ImageNet200 ad the ID data are presented in Table 2. We first analyze the experimental Results of the g-LOND algorithm in terms of the practical metrics. As the Table 1 shown, the g-LOND dramatically improves the detection performance in terms

---

[7]In the code of (Zhang et al., 2023b), OOD is set to be positive.

*Table 2.* The experimental results on **ImageNet200** as ID data. We use ResNet50 as the backbone for each method. The best results are highlighted in **bold**.

| Data | SSB_hard | | | NINCO | | | iNaturalist | | | OpenImage_O | | | Texture | | |
|---|---|---|---|---|---|---|---|---|---|---|---|---|---|---|---|
| Practical | TPR | FPR | F1 | TPR | FPR | F1 | TPR | FPR | F1 | TPR | FPR | F1 | TPR | FPR | F1 |
| ASH | 95.53 | 86.71 | 28.62 | 95.53 | 83.21 | 76.46 | 95.53 | 58.04 | 73.53 | 95.53 | 49.53 | 85.33 | 95.53 | 68.36 | 60.48 |
| Cider | 95.06 | 78.43 | 30.56 | 95.06 | 62.92 | 80.50 | 95.06 | 26.16 | 84.82 | 95.06 | 22.58 | 90.40 | 95.06 | 43.81 | 69.81 |
| Energy | 95.54 | 72.07 | 32.50 | 95.54 | 59.36 | 77.43 | 95.54 | 45.69 | 77.58 | 95.54 | 41.83 | 80.04 | 95.54 | 54.43 | 65.55 |
| KLM | 93.28 | 67.06 | 37.87 | 93.28 | 53.44 | 75.94 | 93.28 | 41.49 | 78.13 | 93.28 | 39.86 | 86.73 | 93.28 | 49.53 | 67.63 |
| MSP | 95.36 | 70.78 | 32.84 | 95.36 | 62.35 | 80.78 | 95.36 | 40.85 | 79.22 | 95.36 | 47.69 | 85.64 | 95.36 | 53.66 | 65.77 |
| RankFeat | 94.68 | 71.07 | 27.42 | 94.68 | 70.54 | 74.60 | 94.68 | 50.85 | 64.06 | 94.68 | 55.59 | 75.91 | 94.68 | 56.70 | 54.50 |
| SHE | 95.44 | 70.71 | 32.89 | 95.44 | 65.94 | 80.03 | 95.44 | 45.46 | 77.62 | 95.44 | 41.57 | 87.05 | 95.44 | 55.60 | 65.04 |
| VIM | 94.75 | 80.27 | 29.99 | 94.75 | 66.44 | 79.57 | 94.75 | 33.47 | 81.71 | 94.75 | 32.15 | 83.94 | 94.75 | 47.30 | 68.13 |
| DICE | 94.80 | 72.11 | 32.28 | 94.80 | 64.34 | 80.06 | 94.80 | 45.36 | 77.35 | 94.80 | 35.21 | 88.19 | 94.80 | 53.80 | 65.46 |
| PALM | **95.69** | 71.90 | 32.60 | **95.69** | 63.13 | 80.77 | **95.69** | 38.36 | 80.31 | **95.69** | 45.50 | 86.29 | **95.69** | 53.83 | 65.86 |
| Gram | 94.11 | 87.20 | 28.14 | 94.11 | 76.93 | 77.03 | 94.11 | 87.76 | 64.55 | 94.11 | 53.69 | 83.70 | 94.11 | 79.58 | 56.28 |
| KNN | 94.79 | 70.14 | 32.88 | 94.79 | 57.09 | 81.69 | 94.79 | 25.59 | 84.93 | 94.79 | 24.73 | 90.19 | 94.79 | 42.82 | 70.14 |
| ODIN | 95.00 | 76.77 | 31.00 | 95.00 | 63.91 | 80.25 | 95.00 | 33.19 | 81.94 | 95.00 | 30.56 | 87.11 | 95.00 | 50.18 | 67.02 |
| g-LOND(ours) | 92.48 | **53.72** | **49.31** | 92.89 | **45.28** | **85.56** | 93.07 | **20.38** | **86.73** | 93.29 | **17.38** | **91.26** | 93.76 | **39.16** | **74.86** |

| Classical | FPR95 | AUC | AUPR | FPR95 | AUC | AUPR | FPR95 | AUC | AUPR | FPR95 | AUC | AUPR | FPR95 | AUC | AUPR |
|---|---|---|---|---|---|---|---|---|---|---|---|---|---|---|---|
| ASH | 68.86 | 79.52 | 42.16 | 67.52 | 85.24 | 89.10 | 29.53 | 95.10 | 94.79 | 34.74 | 91.77 | 95.61 | 40.95 | 91.02 | 87.79 |
| Cider | 79.52 | 77.94 | 39.06 | 68.48 | 86.10 | 91.32 | 45.99 | 92.87 | 93.81 | 35.20 | 92.36 | 95.21 | 56.57 | 90.22 | 87.9 |
| Energy | 73.01 | 79.83 | 43.78 | 65.70 | 85.17 | 90.08 | 44.31 | 92.55 | 93.28 | 41.06 | 90.79 | 93.78 | 56.64 | 89.23 | 85.91 |
| KLM | 74.61 | 77.56 | 37.28 | 71.11 | 83.96 | 88.70 | 46.65 | 91.80 | 91.90 | 67.66 | 86.13 | 90.46 | 64.37 | 87.66 | 82.42 |
| MSP | 72.88 | 80.38 | **46.66** | 64.48 | 86.29 | 91.26 | 41.94 | 92.8 | 93.44 | 49.94 | 88.36 | 92.36 | 56.98 | 89.24 | 86.19 |
| RankFeat | 91.50 | 58.74 | 20.52 | 92.07 | 55.10 | 64.34 | 98.26 | 33.08 | 37.49 | 97.67 | 29.10 | 50.19 | 92.44 | 52.48 | 37.26 |
| SHE | 71.07 | 78.30 | 41.10 | 67.47 | 82.07 | 87.40 | 42.95 | 91.43 | 91.51 | 38.20 | 90.51 | 93.41 | 55.58 | 87.49 | 82.39 |
| VIM | 85.44 | 74.04 | 39.39 | 75.67 | 83.32 | 89.81 | 56.03 | 90.96 | 92.49 | 38.85 | 92.61 | **96.94** | 64.64 | 88.20 | 86.32 |
| DICE | 72.89 | 79.06 | 42.6 | 65.29 | 84.49 | 89.53 | 46.53 | 91.81 | 92.42 | 35.89 | 91.53 | 94.11 | 54.75 | 89.06 | 85.32 |
| PALM | 72.57 | 80.75 | 44.19 | 64.25 | 86.38 | 91.03 | 39.21 | 93.7 | 94.38 | 43.18 | 90.25 | 93.41 | 55.34 | 90.13 | 87.38 |
| Gram | 89.25 | 65.95 | 26.50 | 79.45 | 69.40 | 74.67 | 89.88 | 65.30 | 62.30 | 57.01 | 80.54 | 83.35 | 81.78 | 67.72 | 51.52 |
| KNN | 77.09 | 77.03 | 38.57 | 63.83 | 86.10 | 91.07 | 34.01 | 93.99 | 94.45 | 39.44 | 91.29 | 92.19 | 52.35 | 90.19 | 87.54 |
| ODIN | 76.84 | 77.19 | 39.81 | 63.90 | 83.34 | 88.02 | 33.33 | 94.37 | 94.7 | 40.35 | 90.65 | 93.78 | 50.33 | 90.11 | 86.18 |
| g-LOND(ours) | **63.49** | **80.61** | 43.27 | **54.43** | **86.59** | **91.59** | **20.88** | **96.72** | 95.44 | **33.97** | **92.83** | 94.89 | **37.98** | **91.88** | **88.69** |

of FPR and F1-score for all OOD data, despite a slight decrease in TPR. For example, using CIFAR-100 as ID data and Place365 as OOD data, our proposed method reduces the FPR from $45.63\%$ to $30.62\%$, and improves the F1-score from $42.85\%$ to $52.87\%$ compared with the best baseline, a direct improvement of **15.01%** and **10.02%** at the cost of less than $3\%$ decrease in TPR. Moreover, this improvement still exists in Table 2 in terms of the FPR and F1-score. For instance, using ImageNet200 as ID data and SSB_hard as OOD data, our proposed method reduces the FPR from $67.06\%$ to $53.72\%$, and improves the F1-score from $37.87\%$ to $49.31\%$ compared with the best baseline, a direct improvement of **13.34%** and **11.44%** at the cost of $3.21\%$ decrease in TPR. These experimental results indicate that the g-LOND tends to classify more testing example as OOD while controlling the number of falsely classifying the ID as the OOD. Hence, our method achieves the smaller FPR and the larger F1-score than baselines. The above analysis indicates that our method achieves a better tradeoff between TPR and FPR, which is consistent with our motivation of controlling FDR.

Then, we analyze the results of our method in terms of the classical metrics. From the Tables 2, we find that compared to the baselines, the AUROC and AUPR of the g-LOND algorithm achieve a certain degree of improvement. What's even more exciting, the FPR95 of g-LOND is reduced obviously. For example, using the ImageNet200 as the ID data

and the NINCO as the OOD data, our proposed g-LOND outperforms the best baseline by **9.4%** in terms of FPR95 and by **0.21%** in terms of AUROC. This conclusion still holds for the different OOD data in Table 1. Overall, our proposed g-LOND algorithm achieves promising OOD detection performance.

# 7. Conclusion

In this paper, we systematically investigate OOD detection problem from statistical perspective. Concretely, we formulate the OOD detection task as a online multiple hypothesis testing problem and propose a novel g-LOND algorithm to solve it. Theoretically, the g-LOND algorithm controls FDR for dependent p-values. Besides, we derive the asymptotic FPR of the g-LOND algorithm under the generalized Gaussian-like distribution family, indicating that the FPR of g-LOND tends to $0$ in probability. Finally, the extensive experiments demonstrate the effectiveness of g-LOND algorithm.

# Acknowledgment

This work is supported by the National Natural Science Foundation of China under Grant 624B2106, the Key R&D Program of Hubei Province under Grant 2024BAB038, the National Key R&D Program of China under Grant 2023YFC3604702, the Fundamental Research Funds for

the Central Universities under Grant 2042025kf0045.

## Impact Statement

To our best knowledge, this work has no negative social impact. This work mainly provides a solid theoretical support for the field of the OOD detection. Hence, our work may promote the development of the related applications.

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

# A. Proofs

### A.1. Proof of Theorems 4.4

*Proof.* We will prove these two theorems in three cases.

**Case 1:**  p-values are mutually independent. Note that

$$\text{FDR}(n) = \mathbb{E}\left[\frac{|\mathcal{R}(n) \cap \mathcal{H}_0(n)|}{\max\{1, |\mathcal{R}(n)|\}}\right] = \mathbb{E}\left[\sum_{i=1}^{n} \frac{\mathbf{1}(f(p_i) \leq \alpha_i, i \in \mathcal{H}_0)}{\max\{D(n), 1\}}\right]$$

$$= \sum_{i=1}^{n} \mathbb{E}\left[\frac{\mathbf{1}(f(p_i) \leq \alpha_i, i \in \mathcal{H}_0)}{\alpha_i} \cdot \frac{\alpha_i}{\max\{D(n), 1\}}\cdot\right]$$

According to the definition of $D(n)$, $\max\{D(n), 1\} \geq D(i)$ for $i \in [n]$. Then, we obtain

$$\frac{\alpha}{\max\{D(n), 1\}} \leq \frac{\alpha_i}{D(i-1)+1}$$
$$= \frac{\alpha\gamma_i(D(i-1)+1)}{D(i-1)+1} = \alpha\gamma_i.$$

If $f(\cdot) \in \mathcal{F}_1$, then for any p-value $p_i \leq f(p_i)$. Hence, we have

$$\mathbb{E}(\mathbf{1}(f(p_i) \leq \alpha_i, i \in \mathcal{H}_0)|D(i-1)) \leq \mathbb{E}(\mathbf{1}(p_i \leq \alpha_i, i \in \mathcal{H}_0)|D(i-1))$$
$$\leq \mathbb{E}(\alpha_i|D(i-1)) = \alpha_i.$$

It follows that

$$\mathbb{E}\left[\frac{\mathbf{1}(f(p_i) \leq \alpha_i, i \in \mathcal{H}_0)}{\alpha_i}\right] = \mathbb{E}\left[\mathbb{E}\left(\frac{\mathbf{1}(p_i \leq \alpha_i, i \in \mathcal{H}_0)}{\alpha_i}|D(i-1)\right)\right]$$
$$\leq \mathbb{E}(1|D(i-1)) = 1.$$

The above analysis demonstrates that

$$\text{FDR}(n) \leq \sum_{i=1}^{n} \mathbb{E}(1 \cdot \alpha\gamma_i) \leq \alpha \cdot \sum_{i=1}^{\infty} \gamma_i = \alpha.$$

**Case 2:**  p-values are PRDS.

Recall that $f(\cdot)$ is strictly increasing. By Lemma 4.3, we know that $f(p_1), f(p_2), \cdots, f(p_n)$ are PRDS. The proof in case 1 shows that

$$\text{FDR}(n) \leq \sum_{i \in \mathcal{H}_0(n)} \alpha\gamma_i \cdot \mathbb{E}\left[\frac{\mathbf{1}(f(p_i) \leq \alpha_i)}{\alpha_i}\right].$$

Without loss of generality, we assume $\alpha_i \in (0, 1)$. For a positive number $\epsilon \in (0, 1)$, we choose a positive integer $m$ such that $\alpha_i > \epsilon^m$. Denote $s_j = \epsilon^{m+1-j}$ for $j \in [m+1]$. Note that

$$\mathbb{P}\left(f(p_i) \leq \alpha_i\right) = \mathbb{P}\left(f(p_i) \leq \alpha_i, \alpha_i \in \cup_{j=1}^{m}(s_j, \ s_{j+1}]\right).$$

Then, for $i \in \mathcal{H}_0$, the following chain of inequities hold:

$$\mathbb{E}\left[\frac{\mathbf{1}(f(p_i) \leq \alpha_i)}{\alpha_i}\right] \leq \sum_{j=1}^{m} \frac{\mathbb{P}(f(p_i) \leq s_{j+1}, \alpha_i \in (s_j, \ s_{j+1}])}{s_j}$$

$$= \sum_{j=1}^{m} \frac{\mathbb{P}(f(p_i) \leq s_{j+1}) \cdot \mathbb{P}(\alpha_i \in (s_j, \ s_{j+1}] | f(p_i) \leq s_{j+1})}{\mathbb{P}(f(p_i) \leq s_{j+1})} \cdot \frac{\mathbb{P}(f(p_i) \leq s_{j+1})}{s_j}$$

$$\leq \sum_{j=1}^{m} P(\alpha_i \in (s_j, \ s_{j+1}] | f(p_i) \leq s_{j+1}) \cdot \frac{s_{j+1}}{s_j}$$

$$\leq \epsilon^{-1} \sum_{j=1}^{m} \left(\mathbb{P}(\alpha_i \leq s_{j+1} | f(p_i) \leq s_{j+1}) - \mathbb{P}(\alpha_i \leq s_j | f(p_i) \leq s_{j+1})\right)$$

$$= \epsilon^{-1} \sum_{j=1}^{m-1} \left(\mathbb{P}(\alpha_i \leq s_{j+1} | f(p_i) \leq s_{j+1}) - \mathbb{P}(\alpha_i \leq s_{j+1} | f(p_i) \leq s_{j+2})\right)$$

$$+ \epsilon^{-1} \left(\mathbb{P}(\alpha_i \leq s_{m+1} | f(p_i) \leq s_{m+1}) - \mathbb{P}(\alpha_i \leq s_1 | f(p_i) \leq s_2)\right)$$

$$\leq \epsilon^{-1} \cdot \mathbb{P}(\alpha_i \leq s_{m+1} | f(p_i) \leq s_{m+1}) \qquad \text{(by Lemma 4.3)}$$

$$\leq \epsilon^{-1}.$$

Letting $\epsilon \to 1$ and applying the monotone convergence theorem, we have

$$\mathbb{E}\left[\frac{\mathbf{1}(f(p_i) \leq \alpha_i)}{\alpha_i}\right] \leq 1.$$

It follows that

$$\text{FDR}(n) \leq \sum_{i \in \mathcal{H}_0(n)} \alpha \gamma_i \cdot 1 \leq \alpha.$$

$\square$

### A.2. Prrof of Theore 4.5

*Proof.* Note that $D(n) \geq D(i-1) + \mathbf{1}(f(p_i) \leq \alpha_i)$ and

$$\mathbf{1}(f(p_i) \leq \alpha_i) = \mathbf{1}\left(1 \leq \frac{\alpha_i}{f(p_i)}\right) \leq \frac{\alpha_i}{f(p_i)}.$$

Then, we have

$$\text{FDR}(n) \leq \sum_{i \leq n, i \in \mathcal{H}_0} \mathbb{E}\left[\frac{\mathbf{1}(f(p_i) \leq \alpha_i)}{\max\{D(n), 1\}}\right]$$

$$= \sum_{i \leq n, i \in \mathcal{H}_0} \mathbb{E}\left[\frac{\mathbf{1}(f(p_i) \leq \alpha_i)}{\max\{D(n), 1\}} \cdot \mathbf{1}(f(p_i) \leq \alpha_i)\right]$$

$$\leq \sum_{i \leq n, i \in \mathcal{H}_0} \mathbb{E}\left[\frac{\alpha_i / f(p_i)}{\max\{D(n), 1\}} \cdot \mathbf{1}(D(n) \geq D(i-1) + 1)\right]$$

$$\leq \sum_{i \leq n, i \in \mathcal{H}_0} \mathbb{E}\left[\frac{\alpha_i / f(p_i)}{D(i-1) + 1} \cdot \mathbf{1}(D(n) \geq D(i-1) + 1)\right]$$

$$= \sum_{i \leq n, i \in \mathcal{H}_0} \mathbb{E}\left[\frac{\alpha \gamma_i (D(i-1) + 1)}{(D(i-1) + 1) f(p_i)} \cdot \mathbf{1}(D(n) \geq D(i-1) + 1)\right]$$

$$\leq \sum_{i \leq n, i \in \mathcal{H}_0} \mathbb{E}\left[\frac{\alpha \gamma_i}{f(p_i)}\right].$$

Let random variable $U$ be uniformly distributed on $(0, 1)$. For any $t \in (0, 1)$, we have

$$\mathbb{P}(p_i \leq t) \leq t = \mathbb{P}(U \leq t).$$

If $t \geq 1$, then $\mathbb{P}(p_i \leq t) = \mathbb{P}(U \leq t) = 1$. Hence, $\mathbb{P}(p_i \leq t) \leq \mathbb{P}(U \leq t)$ for any $t \geq 0$. Since $f(\cdot)$ is positive and strictly increasing, it follows that

$$\mathbb{P}(\frac{1}{f(p_i)} \geq x) = \mathbb{P}(p_i \leq f^{-1}(\frac{1}{x})) \leq \mathbb{P}(U \leq f^{-1}(\frac{1}{x})) = \mathbb{P}(\frac{1}{f(U)} \geq x)$$

for any $x \geq 0$. According to the following property of expectation:

$$\mathbb{E}(X) = \int_0^\infty x f(x) \, dx = \int_0^\infty \mathbb{P}(X \geq x) \, dx \tag{8}$$

for any non-negative random variable $X$, then we obtain

$$\mathbb{E}(\frac{1}{f(p_i)}) = \int_0^\infty \mathbb{P}(\frac{1}{f(p_i)} \geq t) \, dt \leq \int_0^\infty \mathbb{P}(\frac{1}{f(U)} \geq t) \, dt$$
$$= \mathbb{E}(\frac{1}{f(U)}) = \int_0^1 \frac{1}{f(u)} \cdot 1 \, du \leq 1.$$

Therefore, we obtain

$$\mathrm{FDR}(n) \leq \sum_{i \leq n, i \in \mathcal{H}_0} \mathbb{E}\left[\frac{\alpha \gamma_i}{f(p_i)}\right] \leq \sum_{i \leq n, i \in \mathcal{H}_0} \alpha \gamma_i \cdot \mathbb{E}(\frac{1}{f(p_i)})$$
$$\leq \sum_{i \leq n, i \in \mathcal{H}_0} \alpha \gamma_i \leq \alpha.$$

$\square$

### A.3. Proof of Theorem 5.2

*Proof.* For notational convenience, we denote by $X_1^{test}, X_2^{test}, \cdots, X_{n_1}^{test}$ the OOD examples in testing set. For $f$ in the g-LOND algorithm, we have $f(x) \geq x$ and $f^{-1}(x) \leq x$. Besides, denote by $\Phi = 1 - \Psi$ the cumulative distribution function (CDF) of testing statistic $T$ under the null hypothesis. For simplicity, we assume $\Phi(0) = 1/2$. Denote by $F_1(t)$ the CDF of the p-values under alternative hypothesis. Then, $F(t)$ can be expressed as

$$F_1(t) = \Psi(\mu - \Phi^{-1}(1 - t))$$

where $\Phi^{-1}$ is the inverse function of $\Phi$. Under the null hypothesis, the CDF of the p-value $F_0(t) = t$. Then the CDF of the p-values can be expressed as

$$F(t) = (1 - \varepsilon)F_0 + \varepsilon F_1(t) = (1 - \varepsilon)t + \varepsilon\Psi(\mu - \Phi^{-1}(1 - t)).$$

Denote $\bar{\Psi} = 1 - F_1$ and $\kappa := \Phi^{-1}(1 - t)$. Because $\Psi \in \mathcal{F}$, we have

$$\kappa = \Phi^{-1}(1 - t) \sim (\lambda \log(1/t))^{1/\lambda}.$$

According to the definition of g-LOND algorithm, for $k \geq 0$, and $m \geq \tau_k + 1$, we have

$$\mathbb{P}(\tau_{k+1} > m \mid \tau_k) = \prod_{i=\tau_k+1}^m \mathbb{P}(f(p_i) > (k+1)\gamma_i) = \prod_{i=\tau_k+1}^m (1 - F(f^{-1}((k+1)\gamma_i)))$$
$$\leq \exp\{-\sum_{i=\tau_k+1}^m F(f^{-1}((k+1)\gamma_i))\}.$$

Denote $\tilde{\tau}_k = \tau_k \wedge n_1$. For $\tau_k < n_1$, we have

$$
\begin{aligned}
\mathbb{E}(\tilde{\tau}_{k+1} \mid \tilde{\tau}_k) &= \tau_k + 1 + \sum_{m=\tau_k+1}^{n_1} \mathbb{P}(\tau_{k+1} > m \mid \tau_k) \\
&\le \tau_k + 1 + \sum_{m=\tau_k+1}^{n_1} \exp\{- \sum_{i=\tau_k+1}^{m} F(f^{-1}((k+1)\gamma_i))\} \\
&= \tilde{\tau}_k + 1 + \sum_{m=\tilde{\tau}_k+1}^{n_1} \exp\{- \sum_{i=\tilde{\tau}_k+1}^{m} F(f^{-1}((k+1)\gamma_i))\}.
\end{aligned}
\tag{9}
$$

Define $t^*$ satisfying $\Phi^{-1}(1 - f^{-1}(t^*)) = \mu$. Note that $\Psi \in \mathcal{F}$. Then, for $t \ge t^*$, we have

$$
F_1(t) = \Psi(\mu - \Phi^{-1}(1 - f^{-1}(t))) \ge \Psi(\mu - \Phi^{-1}(1 - f^{-1}(t^*))) = 1/2.
$$

Define $M_1 := \lfloor ((k+1)C/t^*)^{1/v} \rfloor$. If $i \le M_1$, we have $F_1(f^{-1}((k+1)\gamma_i)) \ge 1/2$.

Without loss of generality, we assume $M_1 \ge n$. Since $\tilde{\tau}_k < n_1 \le M_1$, we have that

$$
\begin{aligned}
\sum_{m=\tilde{\tau}_k+1}^{n_1} \exp\{- \sum_{i=\tilde{\tau}_k+1}^{m} F(f^{-1}((k+1)\gamma_i))\} &\le \sum_{m=\tilde{\tau}_k+1}^{n_1} \exp\{-(m-\tilde{\tau}_k)\varepsilon/2\} \\
&\le \sum_{m=1}^{n_1-\tilde{\tau}_k} \exp\{-m\varepsilon/2\} < \frac{2}{\varepsilon} = 2n_1^\beta.
\end{aligned}
$$

It follows that

$$
\mathbb{E}(\tilde{\tau}_{k+1} \mid \tilde{\tau}_k) \le \tilde{\tau}_k + 1 + 2n_1^\beta,
$$

Therefore, we have

$$
\mathbb{E}(\tau_k \wedge n_1) \le 3kn_1^\beta,
\tag{10}
$$

which completes the proof of Theorem 5.2 $\qquad\square$

## A.4. Proof of Theorem 5.3

*Proof.* For any $\delta > 0$, define $M_2 := \lceil n^\beta/\delta \rceil$. Since $0 < \beta < \frac{1}{2}$, for large $n$, we have $\frac{M_2}{n_1} < \delta$ where $n_1 = |\mathcal{H}_1(n)| = n^{1-\beta}$. For $M_2 \le i \le n_1$, define $\theta_i := i\,\delta\,n_1^{-\beta}$, we get

$$
\begin{aligned}
\mathbb{P}(D(i) < \theta_i) &= \mathbb{P}(\tau_{\lceil \theta_i \rceil} > i) \le \mathbb{P}(\tau_{\lceil \theta_i \rceil} \wedge n_1 \ge i) \\
&\le 3 \cdot \frac{\lceil \theta_i \rceil n_1^\beta}{i} < 3 \cdot \frac{(\theta_i + 1)n_1^\beta}{i} \\
&= 3\delta + 3\frac{n_1^\beta}{i} < 6\delta.
\end{aligned}
$$

According to the definition of g-LOND algorithm, we have

$$
\mathbb{E}[\bar{\Psi}(f^{-1}(\alpha_i))] = \mathbb{E}[\bar{\Psi}(f^{-1}(\gamma_i(D(i-1)+1)))] \le \mathbb{E}[\bar{\Psi}(f^{-1}(\gamma_i D(i)))],
$$

Hence, the expectation of FPR for g-LOND satisfies

$$
\mathbb{E}(\text{FPR}) = \frac{1}{n_1} \sum_{i \in \mathcal{H}_1(n)} \mathbb{E}[\bar{\Psi}(f^{-1}(\alpha_i))] \le \frac{1}{n_1} \sum_{i \in \mathcal{H}_1(n)} \mathbb{E}[\bar{\Psi}(f^{-1}(\gamma_i D(i)))].
\tag{11}
$$

For $1 \le i \le M_2$,

$$
\frac{1}{n_1} \sum_{i \le N_2, i \in \mathcal{H}_1(n)} \mathbb{E}[\bar{\Psi}(f^{-1}(\gamma_i D(i)))] \le \frac{M_2}{n_1}.
$$

For $M_2 + 1 \leq i \leq n_1 < n$,

$$\mathbb{E}[\bar{\Psi}(f^{-1}(\gamma_i D(i)))] = \mathbb{E}[\bar{\Psi}f^{-1}((\gamma_i D(i))) \cdot \mathbf{1}(D(i) < \theta_i)] + \mathbb{E}[\bar{\Psi}f^{-1}((\gamma_i D(i))) \cdot \mathbf{1}(D(i) \geq \theta_i)]$$
$$\leq 6\delta + \bar{\Psi}(f^{-1}(\gamma_i \theta_i)) \leq 6\delta + \bar{\Psi}(f^{-1}(\gamma_n \theta_n)).$$

It follows that

$$\frac{1}{n_1} \sum_{i \in \mathcal{H}_1(n)} \mathbb{E}[\bar{\Psi}f^{-1}((\gamma_i D(i)))] \leq \frac{M_2}{n_1} + \frac{n_1 - M_2}{n_1}(6\delta + \bar{\Psi}(f^{-1}(\gamma_n \theta_n)))$$
$$\leq 7\delta + \bar{\Psi}(f^{-1}(\gamma_n \theta_n)) = 7\delta + \Psi(\mu - \kappa_n),$$

where $\kappa_n = \Phi^{-1}(1 - f^{-1}(\gamma_n \theta_n))$. Since $\nu + \beta - 1 < r$ and $f^{-1}(\gamma_n \theta_n) \leq \gamma_n \theta_n$, as $n \to \infty$, then we have

$$\mu - \kappa_n \geq (r^{1/\lambda} - (\nu + \beta - 1)^{1/\lambda})(\lambda \log n)^{1/\lambda} \to \infty$$

and further $\Psi(\mu - \kappa_n) \to 0$ as $n \to \infty$. Then, we have

$$\limsup_{n \to \infty} \mathbb{E}(\mathrm{FPR}) \leq 7\delta. \tag{12}$$

Note that Eq. (12) holds for any $\delta > 0$, thus $\mathbb{E}(\mathrm{FPR}) \to 0$ as $n \to \infty$. By Markov's inequality, for any $t > 0$, we have

$$\mathbb{P}(\mathrm{FPR}_{g-LOND} \geq t) \leq \frac{\mathbb{E}(\mathrm{FPR}_{g-LOND})}{t} \to 0$$

Hence, we conclude that

$$\mathrm{FPR}_{g-LOND} \to 0 \qquad \text{in probability,}$$

$\square$

