# OpenReview forum: "An Online Statistical Framework for Out-of-Distribution Detection"
_ICML.cc/2025/Conference — ICML 2025 poster_

### Official Review · Reviewer_RpMq · 2025-03-10

**Overall Recommendation:** 5

**Summary:**

The paper focuses on the out-of-distribution (OOD) detection task. Unlike previous research that primarily focuses on designing powerful score functions, this paper introduces a novel perspective by framing OOD detection as a online multiple hypothesis testing problem. The authors propose a Generalized LOND (g-LOND) algorithm with both rigorous theoretical guarantees and strong empirical performance. The g-LOND algorithm enables to control the false discovery rate (FDR). Besides, its false positive rate (FPR) converges to zero in probability. Experimental results show that the g-LOND algorithm outperforms traditional threshold-based methods across various OOD detection benchmarks.

## update after rebuttal

I am willing to champion this paper. I have read all reviews and rebuttals. To my best knowledge, this paper may be the first to propose a novel online hypothesis testing framework for OOD detection with a strong theoretical guarantee. Extensive experiments demonstrate the effectiveness of this framework. Besides, the authors also provide the new experiment results in the rebuttal, which further enhance the claims of this paper. So, I recommend the acceptance of the paper.

**Claims And Evidence:**

Yes. The claims made in the submission is supported by many theoretical results and extensive experiments.

**Essential References Not Discussed:**

No, the paper includes essential references. It adequately discusses prior works that are crucial for understanding the context of the key contributions.

**Experimental Designs Or Analyses:**

Yes. The experimental designs and analyses are sound and well-executed, with a clear framework for evaluating the proposed method.

**Methods And Evaluation Criteria:**

Yes. The proposed method is based on the statistical hypothesis testing framework, and the evaluation criteria are appropriate for OOD detection task.

**Other Comments Or Suggestions:**

See weaknesses.

**Other Strengths And Weaknesses:**

Strengths

- The paper introduces a novel perspective on OOD detection by framing it as an online multiple hypothesis testing problem. This departure from traditional approaches adds originality and encourages new thinking in the field.

- The proposed g-LOND algorithm is innovative and theory-inspired, with good interpretability. This methodology represents a departure from conventional threshold-based methods.

- The techniques of the proofs in the Appendix are sound and detailed.

- The proposed method is distribution-free and easy to implement. Besides, extensive experiments are conducted to validate the proposed g-LOND procedure, demonstrating its superiority over traditional methods.

Weaknesses

- In Algorithm 1, the proposed method utilizes a calibrated set for hypothesis testing.  I think a detailed explanation for it would be helpful.

- The authors need to provide a simple discussion of the motivation.

**Questions For Authors:**

In multiple hypothesis testing, do other evaluation criteria exist similar to FDR?

**Relation To Broader Scientific Literature:**

This paper is based on the previous multiple hypothesis testing literature. The authors modify the traditional LOND algorithm and extend its theoretical results such that the proposed method can be applied to OOD detection task.

**Theoretical Claims:**

Yes. I check the proofs of Theorem 4.5, Theorem 4.6 and Theorem 5. These proofs are sound.

---

> ### Author Rebuttal · Authors · 2025-04-01
>
> __W1__. In Algorithm 1, the proposed method utilizes a calibrated set for hypothesis testing. I think a detailed explanation for it would be helpful.
>
> __Ans-W1__. In practice, we just need to randomly sample a small number of examples from the ID training set as the calibrated set, without any special operations. In our experiments, for  CIFAR-100 as ID data, the calibrated set contains 2000 ID examples; for ImageNet-200 as ID data, the calibrated set contains 10000 ID examples.
>
>
> __w2__. The authors need to provide a simple discussion of the motivation.
>
> __Ans-W2__. Usually, the traditional threshold-based decision rule in Eq.(1) performs well on ID data but performs poorly on OOD data. Because the selection of its threshold just considers a high TPR on ID validation set. By contrast, our method considers the performance on both ID data and OOD data simultaneously. Specifically, our method can control FDR. Intuitively, to control FDR, the g-BH tends to reject more null hypotheses while maintaining small false rejections of the null hypothesis. Small false rejections means to control the number of falsely classifying the ID as the OOD (maintaining a high TPR) and more rejection of null hypotheses means to classify more testing examples as OOD (leading to a low FPR on OOD data).
>
> __Q1__. In multiple hypothesis testing, do other evaluation criteria exist similar to FDR?
>
> __Ans-Q1__. Family-wise error rate (FWER) and  marginal FDR (mFDR)  are two similar evaluation metrics. The FWER can be expressed as
> $$ FWER = P(R\cap H _0 > 0). $$
> The mFDR can be expressed as
> $$ mFDR = \frac{E(R\cap H_0)}{E(R)}. $$

---

> > ### Comment · Reviewer_RpMq · 2025-04-02
> >
> > Thank you for your response. After reviewing the rebuttal addressed to me and those for other reviewers, I am willing to maintain my score for acceptance.

---

### Official Review · Reviewer_eu3U · 2025-03-13

**Overall Recommendation:** 2

**Summary:**

this paper thinks the OOD detection task from an perspective of online multiple hypothesis testing.

the g-LOND algorithm controls false discovery rate (FDR) at pre-specified level without the consideration for the dependence between the p-values.

Along with thorecticla analysis, the empirical effectiveness of g-LOND is evluated on cifar-100 and imagenet-200.

**Claims And Evidence:**

yes

**Essential References Not Discussed:**

[a] Out-of-distribution detection based on in-distribution data patterns memorization with modern hopfield energy

[b] LINe: Out-of-Distribution Detection by Leveraging Important Neurons

[c] Extremely simple activation shaping for out-of-distribution detection

[d] NECO: NEural Collapse Based Out-of-distribution detection

[e] Optimal Parameter and Neuron Pruning for Out-of-Distribution Detection

[f] VRA: Variational Rectified Activation for Out-of-distribution Detection

[h] Neuron Activation Coverage: Rethinking Out-of-distribution Detection and Generalization

[i] Tractable Density Estimation for Out-of-Distribution Detection

**Experimental Designs Or Analyses:**

yes

**Methods And Evaluation Criteria:**

yes

**Other Comments Or Suggestions:**

see weakness

**Other Strengths And Weaknesses:**

**Strength**
1. this paper investigates OOD detection from a fresh perspective
2. this paper is well written
3. the evalution, which is based on 6 metrics, is comprehensive

**Weakness**
1. the evaluiation on CIFAR-10 and ImageNet-1k lacks
2. the comparision with the mostly recent baselines [a,b,c,d,e,f,g,h,i] lacks
3. due to the rapid development of neural networks, CLIP-based models should be also considered.

*While I am happy to rasie my rating if the authors can address my concerns, i will also consider reviews from other reviewers regarding the technological nolvety of hypothesis testing used in this paper before making my final decison.*

**Questions For Authors:**

no

**Relation To Broader Scientific Literature:**

N/A

**Theoretical Claims:**

yes

---

> ### Author Rebuttal · Authors · 2025-04-01
>
> __Q1__: about the reference.
>
> __Ans-Q1__: Thank you for providing these meaningful references [a]-[i]. We will discuss these papers in related work section.
>
>
> __W1,2__. the evaluiation on CIFAR-10 and ImageNet-1k lacks. the comparision with the mostly recent baselines [a,b,c,d,e,f,g,h,i] lacks.
>
> __Ans-W1,2__. According to your suggestions, we conduct corresponding experiments on CIFAR-10 and ImageNet-1k using the methods  LINe[b], NECO[d],VAR[f], NAC[h] and CONJNORM[i]. We use the code of OpenOOD [j].
> It should be noted that reference [g] in your Review is missing. Besides, the methods SHE in [a] and ASH in [c] have been used our baselines (see Section 6.1 baselines). Since [e] does not open their code and time of rebuttal period is limited, we do not implement the method in [e]. The experimental results are presented in Tables 2-5 of PDF (see https://anonymous.4open.science/r/gLOND-BBCE/Experimental%20Results%20for%20Rebuttal.pdf ), which demonstrate the superiority of our proposed g-LOND algorithm over these methods in [a][b][c][d][f][h][i].
>
>
>
>
>
> __W3__. Due to the rapid development of neural networks, CLIP-based models should be also considered.
>
> __Ans-W3__. Because of the limitation of time, we just choose four methods based on CLIP architecture as our baselines, including MCM[k], GLMCM[l], SeTAR-MCM and SeTAR-GLMCM[m]. We use ImageNet-1k as ID data, and use iNaturalist, Places, Sun and Texture as the OOD data. Our code is based on [m]. The experimental results are presented in Table 1 of PDF (see https://anonymous.4open.science/r/gLOND-BBCE/Experimental%20Results%20for%20Rebuttal.pdf), which demonstrate the superiority of our proposed g-LOND algorithm over these CLIP-based methods in [k]-[m].
>
>
>
> [a] Out-of-distribution detection based on in-distribution data patterns memorization with modern hopfield energy
>
> [b] LINe: Out-of-Distribution Detection by Leveraging Important Neurons
>
> [c] Extremely simple activation shaping for out-of-distribution detection
>
> [d] NECO: NEural Collapse Based Out-of-distribution detection
>
> [e] Optimal Parameter and Neuron Pruning for Out-of-Distribution Detection
>
> [f] VRA: Variational Rectified Activation for Out-of-distribution Detection
>
> [h] Neuron Activation Coverage: Rethinking Out-of-distribution Detection and Generalization
>
> [i] Tractable Density Estimation for Out-of-Distribution Detection
>
> [j] OpenOOD: Benchmarking Generalized
> Out-of-Distribution Detection
>
> [k] Delving into out-of-distribution detection with vision-language representations.
>
> [l] Zero-shot in-distribution detection in multi-object settings using vision-language foundation models.
>
> [m] SeTAR: Out-of-Distribution Detection with Selective Low-Rank Approximation.

---

### Official Review · Reviewer_UdXp · 2025-03-13

**Overall Recommendation:** 4

**Summary:**

This paper studies the OOD detection task as an online multiple hypothesis testing problem. It presents a new algorithm, called the generalized LOND algorithm (g-LOND), built upon the well-known LOND algorithm. They provide theoretical results about the false discovery rate (FDR) and false positive rate (FPR) for their algorithm. The paper also provides many experiments on OOD detection, comparing to several baselines on several datasets. Overall, the proposed method offers a systematic and theoretically grounded solution to the OOD detection problem, enhancing the reliability of sensitive applications.

**Claims And Evidence:**

Yes. The authors conduct extensive experiments to demonstrate the effectiveness of the g-LOND algorithm.

**Essential References Not Discussed:**

To the best of my knowledge, the essential and relevant references are discussed

**Experimental Designs Or Analyses:**

Yes. The experimental designs and analyses are sound. This paper evaluates the proposed method using practical and classical criteria, making its conclusions convincing.

**Methods And Evaluation Criteria:**

Yes. The proposed g-LOND algorithm makes sense for the OOD detection problem.

**Other Comments Or Suggestions:**

No

**Other Strengths And Weaknesses:**

(1) Overall, this paper is well-motivated and has a clear organization. Besides, its notations and the definitions are clear, and the ideas are easy to follow.

(2) Different from previous literature which mainly focuses on designing or training score function, this paper study the OOD detection problem based hypothesis testing framework and proposes a novel g-LOND algorithm to solve it.

(3) Under some conditions, the authors establish the asymptotic theories about FPR, which  remains underexplored in previous literature.

(4) Extensive experimental results on multiple benchmarks (including large-scale and high-resolution ImageNet) can support the proposed method.

 Weaknesses: I have not identified major weaknesses of this paper, while I do have some minor concerns that are listed in the "Questions For Authors" part.

**Questions For Authors:**

(1) While the paper rigorously develops its theoretical framework, it would be beneficial to outline any underlying assumptions made in this paper.

(2) There are some typos in the paper in Appendix. The authors should check the text.

**Relation To Broader Scientific Literature:**

This paper clearly explains how it is related to previous work. The authors establish the connection between OOD detection and online multiple hypothesis testing, and then proposes novel g-LOND algorithm with strong statistical guarantee.

**Theoretical Claims:**

Yes. I have checked the correctness of the proofs in Appendix, which are both clear and solid.

---

> ### Author Rebuttal · Authors · 2025-04-01
>
> __Q1__. While the paper rigorously develops its theoretical framework, it would be beneficial to outline any underlying assumptions made in this paper.
>
> __Ans-Q1__. In Theorem 4.5 and 4.6, we have no underlying assumptions. In Theorem 5.3, we just assume that the testing statistic follows generalized Gaussian-like distribution family in Definition 5.1.
>
>
> __Q2__.  There are some typos in the paper in Appendix. The authors should check the text.
>
> __Ans-Q2__. Thanks for your careful review. We have checked the typos in Appendix. We will fix these typos in new version.

---

> > ### Comment · Reviewer_UdXp · 2025-04-02
> >
> > My questions have been addressed. Thanks for the reply.

---

### Official Review · Reviewer_sP8g · 2025-03-17

**Overall Recommendation:** 2

**Summary:**

This work investigates out-of-distribution (OOD) detection from the perspective of online multiple hypothesis testing. This paper proposes a generalized LOND algorithm that controls the false discovery rate even under dependent p-values. This work also derives the asymptotic false positive rate of the g-LOND algorithm under a generalized Gaussian-like distribution family. Experiments demonstrate the effectiveness of g-LOND.

**Claims And Evidence:**

The main claim that the generalized LOND algorithm controls the false discovery rate even under dependent p-values is supported by formal theorem and detailed proofs in the appendix. Furthermore, extensive experiments demonstrate its empirical improvements.

**Essential References Not Discussed:**

A few works appear essential for understanding the paper’s key contributions but are not currently discussed in the paper. For example:

[1] GEN: Pushing the Limits of Softmax-Based Out-of-Distribution Detection, CVPR 2023

[2] POEM: Out-of-Distribution Detection with Posterior Sampling, ICML 2022

[3] How Does Unlabeled Data Provably Help Out-of-Distribution Detection? ICLR 2024

**Experimental Designs Or Analyses:**

The experimental design, which uses commonly accepted OOD datasets like SVHN, Places365, and iNaturalist, supports the paper’s claims. The authors rely on ResNet18 and ResNet50 as backbone models. However, using more advanced architectures such as CLIP could support a broader analysis.

**Methods And Evaluation Criteria:**

The proposed method frames OOD detection as an online multiple hypothesis testing problem, leveraging FDR-control procedures to handle dependence among p-values, which is a reasonable approach for OOD tasks. The use of public benchmarks such as TinyImageNet and SVHN is standard, and TPR, FPR, and F1 provides appropriate metrics for evaluation.

**Other Comments Or Suggestions:**

None

**Other Strengths And Weaknesses:**

Strengths

1. The paper leverages FDR control and provides a theoretical analysis of the false positive rate.

2. Extensive experiments on several public benchmarks demonstrate the method’s effectiveness.

Weaknesses

1. The proposed method may be sensitive to both the size and quality of the calibration set.

2. Evaluation is limited to ResNet models; exploring and validating the approach on more advanced architectures (e.g., Transformers, CLIP) would be beneficial.

**Questions For Authors:**

Refer to the Weaknesses section for the detailed questions.

**Relation To Broader Scientific Literature:**

This work connects the existing statistical framework of LOND to OOD detection, which is an online multiple hypothesis testing method. This provides a rigorous theoretical underpinning for OOD methods, which are important for ensuring AI safety.

**Theoretical Claims:**

I check the theoretical claims and proofs, which looks logically consistent. The generalized Gaussian assumption may not match the complexity of real-world data.

---

> ### Author Rebuttal · Authors · 2025-04-01
>
> __Q1__: about the reference.
>
> __Ans-Q1__: Thank you for providing these meaningful references [1]-[3]. We will discuss these papers in related work section.
>
>
>
> __Weakness 1__. The proposed method may be sensitive to both the size and quality of the calibration set.
>
> __Ans-w1__. In practice, we just need to randomly sample a small number of examples from the ID training set as the calibrated set, without any special operations. For example, when using CIFAR-10 as ID data, the calibrated set contains 2000 ID examples. According to your comments, we use CIFAR-10 as ID data, and  SVHN and Place365 as OOD data to study the performance of TPR, FPR and F1-score with variant size of calibrated set. The experimental results are presented in Figure 1 of PDF (see https://anonymous.4open.science/r/gLOND-BBCE/Experimental%20Results%20for%20Rebuttal.pdf). The results show that the performance of our method does not significantly vary as the size of the calibration set increases.
>
>
>
>
> __Weakness 2__. Evaluation is limited to ResNet models; exploring and validating the approach on more advanced architectures (e.g., Transformers, CLIP) would be beneficial.
>
> __Ans-w2__. Because of the limitation of time, we just choose four methods based on CLIP architecture as our baselines, including MCM[4], GLMCM[5], SeTAR-MCM and SeTAR-GLMCM[6]. We use ImageNet-1k as ID data, and use iNaturalist, Places, Sun and Texture as the OOD data. Our code is based on [6]. The experimental results are presented in Table 1 of PDF (see https://anonymous.4open.science/r/gLOND-BBCE/Experimental%20Results%20for%20Rebuttal.pdf), which demonstrate the superiority of our proposed g-LOND algorithm over these CLIP-based methods in [4]-[6].
>
>
> [1] GEN: Pushing the Limits of Softmax-Based Out-of-Distribution Detection, CVPR 2023
>
> [2] POEM: Out-of-Distribution Detection with Posterior Sampling, ICML 2022
>
> [3] How Does Unlabeled Data Provably Help Out-of-Distribution Detection? ICLR 2024
>
> [4] Delving into out-of-distribution detection with vision-language representations.
>
> [5] Zero-shot in-distribution detection in multi-object settings using vision-language foundation models.
>
> [6] SeTAR: Out-of-Distribution Detection with Selective Low-Rank Approximation.

---

### Decision · Program_Chairs · 2025-05-01

**Decision:**

Accept (poster)

**Comment:**

After review, the paper received two positive and two negative reviews. Following the author’s rebuttal, all reviewers maintained their original positions. During the AC-reviewer discussion, all reviewers actively participated—one of the negatively inclined reviewers raised concerns about the paper’s experimental setup and evaluation metrics.

Given these points, the AC conducted a thorough assessment of the paper and concluded that its technical contributions outweighed its weaknesses (including evaluation limitations and insufficient discussion of related work). Particularly, after checking the paper, the authors include reasonable baselines and evaluation metrics in their experiments. Also, related works suggested by reviewers are compared in the authors' rebuttal.

Accordingly, the AC recommends acceptance. The authors are encouraged to revise the paper carefully, addressing the reviewers’ feedbacks by providing a more thorough discussion of related work.